

# Genetic characterization of hepatitis B virus genotypes among patients with chronic infection in Sulaimaniyah city, Iraq

Mardin Othman Abdulqadir[1], Peshnyar Muhammad Atta Rashid[2], Ali Hattem Hussain[3], Heshu Sulaiman Rahman[4] and Shahow Abdulrehman Ezzaddin[5]

[1] Medical Laboratory Department, Technical College of Health, Sulaimani Polytechnic University, Republic of Iraq, Sulaimaniyah, Iraq

[2] Department of Medical Laboratory Sciences, Komar University of Science and Technology, Sulaimaniyah, Republic of Iraq, Sulaimaniyah, Iraq

[3] Nursing Department, Technical College of Health, Sulaimani Polytechnic University, Sulaimaniyah, Republic of Iraq, Sulaimaniyah, Iraq

[4] Department of Physiology, College of Medicine, Sulaimani University, Sulaimaniyah, Republic of Iraq, Sulaimaniyah, Iraq

[5] Family and Community Medicine Department, College of Medicine, Sulaimani University, Sulaimaniyah, Republic of Iraq, Sulaimaniyah, Iraq

Corresponding authors
Peshnyar Muhammad Atta Rashid,
Dr.peshnyar@yahoo.com, peshn-yar.atta@komar.edu.iq
Heshu Sulaiman Rahman,
heshu.rhaman@univsul.edu.iq

## ABSTRACT

**Background.** Hepatitis B virus (HBV) genotypes are distributed unevenly throughout the world's regions. The researchers' goal in this study was to find out which HBV genotypes are now prevalent in the blood of chronic HBV patients in Iraq's Kurdistan Region's Sulaimaniyah governorate.

**Methods.** Genotyping was carried out utilizing Polymerase Chain Reaction (PCR) type-specified primers. Thirty-three chronic HBV patients were included in the HBV genotyping assay. Phylogenic trees of Pre-S1/Pre S2/S genes' nucleotide sequences were constructed using 36 HBV isolates.

**Results.** All the patients had HBV genotype D. Additionally, two samples were further analyzed by sequencing and deposited in GenBank as HBV/Sul-1/2021 accession numbers MZ077051 and HBV/Sul-2/2021 accession numbers MZ077052. Phylogenic analysis indicated that the HBV isolates belong to sub-genotype D1/serotype ayw2. The HBV/Sul-2/2021 had two sequence deletion mutations from G61del-T87del, which accounted for 27 amino acid deletions, and ten other mutations were identified in the carboxylic terminus of the pre-S1 from Q104del-R113del. Accordingly, 37 amino acids were deleted in the S promoter region. Several other substitution mutations were recorded in both HBV isolates.

**Conclusion.** Patients with chronic HBV were found to have the HBV sub-genotype D1/subtype ayw2 with no mixed genotypes. HBV/Sul-1/2022, a new strain with a 37-amino acid mutation, was found to be distinct from any previously known HBV isolates.

## INTRODUCTION

HBV infections are common worldwide; it is estimated that over 275 million people are infected with HBV, corresponding to a global prevalence rate of nearly 3.5% (*Razavi, 2020*). Some research in Iraq indicated the prevalence of HBV infection was between 0.7 and 1.37% (*Hussein et al., 2021*; *Othman & Abbas, 2020*). The latest research in Iraq showed that the prevalence of occult HBV in the middle province is in the intermediate zone of endemicity (*Salman, Ali & Hwaid, 2022*).

An HBV infection's outcome is impacted by several variables, including viral load, genotype, mutations, host, and environmental influences (*Iannacone & Guidotti, 2022*). HBV comprises 10 genotypes, from HBV-A to HBV-J, with differences in nucleotides ranging from 7.5 to 15% across whole genomes. Approximately forty sub-genotypes are identified, each of which differs from the total genomic sequence by between 4–7.5% and is given a name consisting of the genotype letter followed by a digit (*Magnius et al., 2020*).

Many researchers have reported a link between different genotypes, the clinical course of infection, disease progression, treatment response, and disease prognosis (*Rybicka et al., 2019*). The pathological impacts of different HBV genotypes are now partially recognized. Genotypes B and C are associated with higher intracellular and extracellular viral DNA than genotypes A and D. In addition, genotype C is associated with high replication capacity, increasing genotype-related liver damage (*Raihan et al., 2019*).

Various genotypes respond differently to antiviral drugs (*Guo et al., 2019*; *Cho & Choe, 2016*). Furthermore, variable HBV genotypes have different rates of liver cirrhosis and hepatocellular cancer development. For example, HBV genotype C has been linked to a higher risk of liver cirrhosis and hepatocellular cancer (*Glebe et al., 2021*). This study was performed in the Sulaimaniyah governorate in the Kurdistan Region of Iraq to track the genotypes currently circulating in the blood of people with chronic HBV infections. Unfortunately, there have been no reports about the HBV genotype in Kurdistan for about a decade since the last report about mixed infection of HBV genotype A+B+C+D in 2013 (*Rashid & Salih, 2015*).

## MATERIAL AND METHODS

### Subjects

This HBV genotyping investigation was performed on 33 chronic HBV adults who visited an outpatient department at Shahid Hadi Consultation Clinic in Sulaimaniyah, Iraq, from January 2020 to March 2021. Only those patients who were over 18 years old were included in this study and tested positive for HBsAg at least six months before the study. The detection of HBsAg is done by the Elecsys® HBsAg II assay kit intended for use on the Cobas e 411 immunoassay analyzer (Roche, Mannheim, Germany). The method relies on an electrochemiluminescence immunoassay principle.

### Sampling

Each patient provided five mL of blood. The blood was drawn aseptically by venipuncture, and the whole blood was collected in plain red-topped tubes. The whole blood was left

 

undisturbed at room temperature for around 15 min to clot. Then, the clot was removed by centrifuging for 10 min at 2,000×g. The resultant supernatant is referred to as serum. The serum was immediately divided into 0.5 mL aliquots and stored and transported at −20 °C. The add prep Viral Nucleic Acid Extraction kit (Add Bio, Gyeongbuk, Republic of Korea) was used to extract hepatitis B viral DNA from the serum samples described by the manufacturer.

## PCR based genotyping

A nested PCR-based genotyping approach was used to identify HBV genotypes A through F from the collected viral DNA (*Raihan et al., 2019*). A universal outside primer was used to amplify 1,074 bp of the Pre-S1 through S genes from all HBV genotypes in the first round of amplification (Table 1). Nested PCR primers were designed using the conserved nucleotides found in the first-round PCR product of HBV DNA amplification. The nested PCR used to distinguish HBV genotypes generates different sizes of amplified DNA. Each sample was subjected to two separate nested PCRs, each in a distinct combination. Genotypes D (119 base pairs), E (167 base pairs), and F (97 base pairs) could be identified using the A mix reaction, whereas the B mix reaction allowed the identification of A (68 base pairs), B (281 base pairs), and C (122 base pairs).

The PCR amplification reaction was done according to the manufacturer's instructions using the Add Star Taq master mix PCR kit (Add Bio, Gyeongbuk, Republic of Korea). In brief, the simplex PCR reaction was completed up to a final volume of 20 µL by DEPC-H$_2$O (3.0 µL), 1.0 µL of 10 pmol for each of the universal P1 forward and S1-2 reverse primers, and 5.0 µL of DNA sample. The thermocycler (ESCO Thermocycler, Singapore) was set up for an initial denaturation phase of 5 min at 95 °C, followed by 40 cycles of denaturation at 94 °C for 30 s, annealing at 50 °C for 30 s, extension at 72 °C for 30 s, and a final extension phase of 5 min at 72 °C.

The nested multiplex PCR amplification reaction was conducted in 0.2 mL tubes using Add Star Taq master mix PCR (Add Bio, Korea). Approximately 1.0 µL of the first-round PCR product was added to both the A and B mixes, and 1.0 µL of 10 pmol of each genotype-specific primer was added, according to *Naito, Hayashi & Abe (2001)* (Table 1). The mixtures were compiled up to a final volume of 20 µL. The nested PCR reaction was carried out for 40 cycles with the following parameters, pre-heated to 94 °C for 5 min, denaturation at 94 °C for 30 s, annealing at 58 °C for 30 s, and extension at 72 °C for 40 s. The final extension was at 72 °C for 5 min.

Finally, the PCR products were examined by loading 6.0 µL of PCR product on a 2% agarose gel in 1× TBE buffer. The gel was stained with ten µL safe gel dye (Add Bio). Electrophoresis was run at 120 volts for an hour on the electrophoresis system. The 50 bp DNA ladder migration pattern was used to look at the amplicons of the PCR product (Fig. 1).

## Sequencing and phylogenic analysis

The outer PCR products (first round) of two samples have been subjected to Sanger sequencing in the Macrogen sequencing facility in South Korea. The nucleotide identity
**Table 1  List of primer sequence and specificity (*Naito, Hayashi & Abe, 2001*).**

| Primer name | Direction | | Sequences 5′–3′ | Identification of genotypes | Amplicons bp |
|---|---|---|---|---|---|
| P1b | forward | First round PCR | TCA CCA TAT TCT TGG GAA CAA GA | All genotypes | 1,063 |
| S1-2 | reverse | | CGA ACC ACT GAA CAA ATG GC | | |
| | | | **Nested PCR reactions** | | |
| B2 | Common forward | Mix A | GGC TCM AGT TCM GGA ACA GT | Forward primer type A-C | |
| BA1R | Reverse | | CTC GCG GAG AT GAC GAG ATG T | Genotype A | 68 |
| BB1R | Reverse | | CAG GTT GGT GAG TGA CTG GAG A | Genotype B | 281 |
| BC1R | Reverse | | GGT CCT AGG AAT CCT GAT GTT G | Genotype C | 122 |
| B2R | Common reverse | Mix B | GGA GGC GGA TYT GCT GGC AA | Reverse primer type D-F | |
| BD1 | forward | | GCC AAC AAG GTA GGA GCT | Genotype D | 119 |
| BE1 | forward | | CAC CAG AAA TCC AGA TTG GGA CCA | Genotype E | 167 |
| BF1 | forward | | GYT ACG GTC AGG GT TAC CA | Genotype F | 97 |

**Notes.**

M, Characterizes a nucleotide that could be either an A or a C; Y, Characterizes a nucleotide that could be a C or a T.

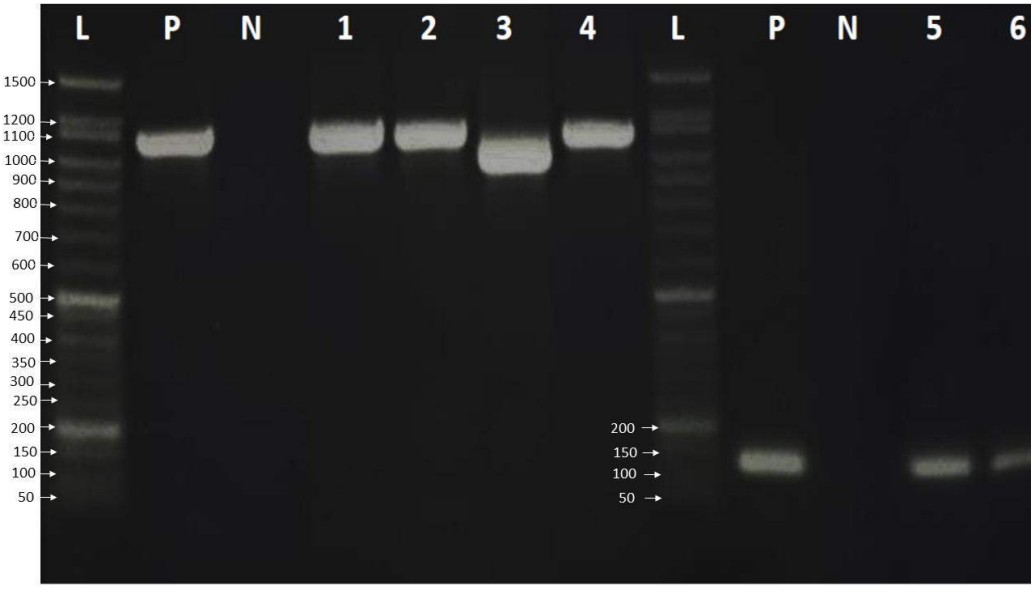

**Figure 1  Agarose gel electrophoresis pattern shows PCR amplification of HBV genome and HBV genotype D.** Lane L: 50 bp DNA ladder, Lanes 1–4: HBV amplicon about 1,063 bp of PreS, PreS2 and S gene, Lane 3 showed HBV deletion mutation of 111 nucleotides. Lanes 5-7: showed 119 bp amplification of HBV genotype D, lane P: positive control, which was already verified by sequencing, and lane N: negative control.

was confirmed by sequencing both ends of the amplicon. Mega 10 program was used to perform multiple sequence alignment of amino acid HBsAg (Pre-S1, Pre-S2, and large S) gene of Sulaimaniyah HBV isolates. It was compared to the peptide sequences of other HBsAg with the greatest percentage of identities and reference HBV isolates retrieved from NCBI. Phylogenic trees were constructed using 36 HBV strains' Pre-S1/Pre-S2/S

genes' nucleotide sequences. The NCBI reference sequences for HBV genotypes and HBV D sub-genotypes were obtained from the NCBI Gene Bank. The Clustal W technique performed multiple alignments of these sequences (*Thompson, Higgins & Gibson, 1994*). A neighbor-joining phylogenetic analysis was performed using MEGA 10, and bootstrap values were calculated using 1000 replicates of the original data (*Kumar et al., 2018*).

## Ethical approval

All procedures performed in this study followed the ethical standards of the national research committee and the 1964 Helsinki declaration and its later amendments or comparable ethical standards. On the other hand, written informed consent was obtained from the patients to publish this data. Therefore, the scientific and ethical committee approved the research of the Kurdistan Institute for Strategic Study and Scientific Research, Sulaimaniyah, Iraq (MLD38-2020).

## RESULTS

### HBV detection and genotyping

A total of 33 chronic hepatitis B patients were examined by PCR assay using a universal primer pair to detect all HBV genotypes. All samples were positive and gave the expected amplicons (about 1,063 bp). However, one sample had a smaller amplicon indicating a deletion mutation in the genomic sequence (Fig. 1) which was then verified by DNA sequencing. The positive samples were genotyped using type-specific primers (Table 1), and all patients had HBV genotype D according to the migratory pattern of gel electrophoresis.

### HBV sequencing

Sanger sequencing on both ends of the outer PCR amplicons validated the two samples' nucleotides. The HBV/Sul-1/2021 accession number MZ077051 and HBV/Sul-2/2021 accession number MZ077052 were then deposited to GenBank. A total of 37 amino acids were deleted in the HBV/Sul-2/2021 strain attributed to two sequence deletion mutations; the first mutation occurred at G61del-T87del, which is about 27 amino acid deletions, while the second mutation occurred at Q104del-R113del, which accounts for ten amino acid deletions. As it has not been reported earlier, the HBV/Sul-2/2021 may be a unique nucleotide sequence (Fig. 2).

### Phylogenic analysis

Phylogenetic analysis was done for a fragment of 1,063 bp of partial Pre-S1/Pre-S2/S genomic region of the current isolates of HBV with the reference HBV sequences in the GenBank. The phylogenetic analysis grouped HBV sequences into eight genotypes (A-H) and ten sub-genotypes (D1-D10) of HBV genotype D (Fig. 3). HBV/Sul-1/2021 and HBV/Sul-2/2021 both belonged to sub-genotype D1 based on the tree's topology.

### HBV serotype

The amino acid sequence of the HBV S protein was deduced to perform the HBV serotyping analysis. Based on Arg122 and Lys160, Pro127, Gly159, and Thr140 in the deduced amino acid sequences of the S gene, all observed Sulaimaniyah HBV serotypes in this study had been categorized as belonging to the serotype ayw2 (Fig. 2).
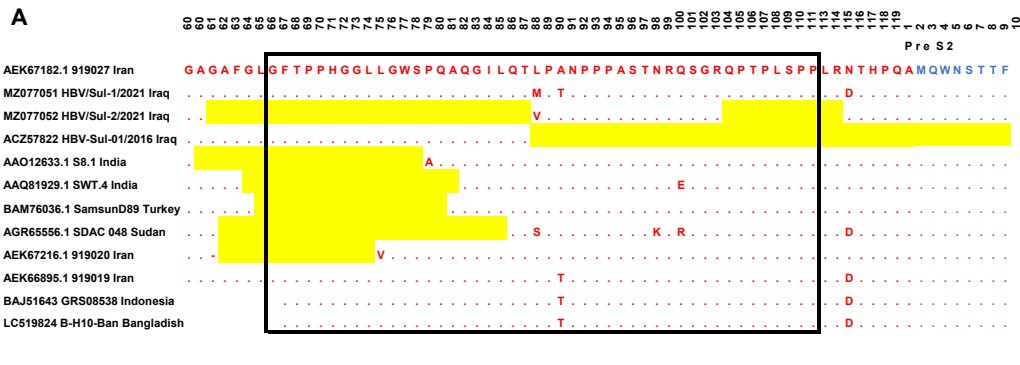

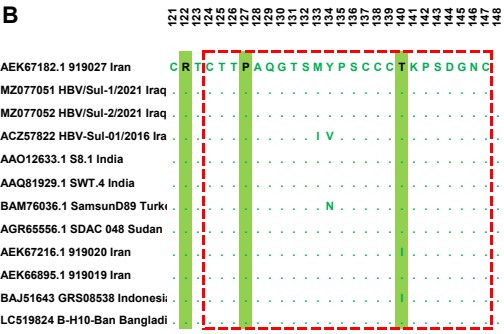

**Figure 2 Multiple sequence alignment of Sulaimani HBV isolates with partial Pres1/preS2/S amino acid of HBV genotype D1 isolates available in GenBank.** PreS1 region is indicated in red color, PreS2 is indicated in Blue, and the S region is indicated in green color. The mutation deletion is highlighted in yellow. The S-promoter region is indicated in a black box. The (A) determinant region is indicated in the red dashed line box. (B) The serotype determinant amino acids are highlighted in green.

## Genetic analysis of the Pre-S1/Pre-S2 region

In the ongoing study, two isolates of Sulaimaniyah HBV had similar 11 amino acid deletions at the beginning of the Pre-S1 region as a categorization of genotype D. In HBV/Sul-2/2021, additional long-stretch deletions were observed in the Pre-S1 region from amino acid residues G61del-T87del, as well as ten other mutations were identified in the carboxylic terminus of the Pre-S1 from Q104del-R113del. Accordingly, 37 amino acids of Pre-S1 protein were deleted, which overlaps 134 nucleotide deletions in the S promoter region (Fig. 2). HBV/Sul-1/2021 had no deletion mutations in Pre-S1/Pre-S2/S amino acids. However, there were various missense point mutations in both isolates of the current study. HBV/Sul-1/2021 had M88L and T90A in the middle of the Pre-S1 protein and D114N at the carboxylic end of the Pre-S1 part. In addition, HBV/Sul-2/2021 had T26P, V55A, and V88L substitutions in the Pre-S1 region. There was no amino acid mutation in the hepatocellular binding site of the virus located at amino acids 21–46, and no frameshift mutation was detected.

## Genetic analysis of the S region

The current Sulaimaniyah HBV strain does not have a premature stop codon mutation in the S gene. But one of the Sulaimaniyah HBV strains already found has an early stop

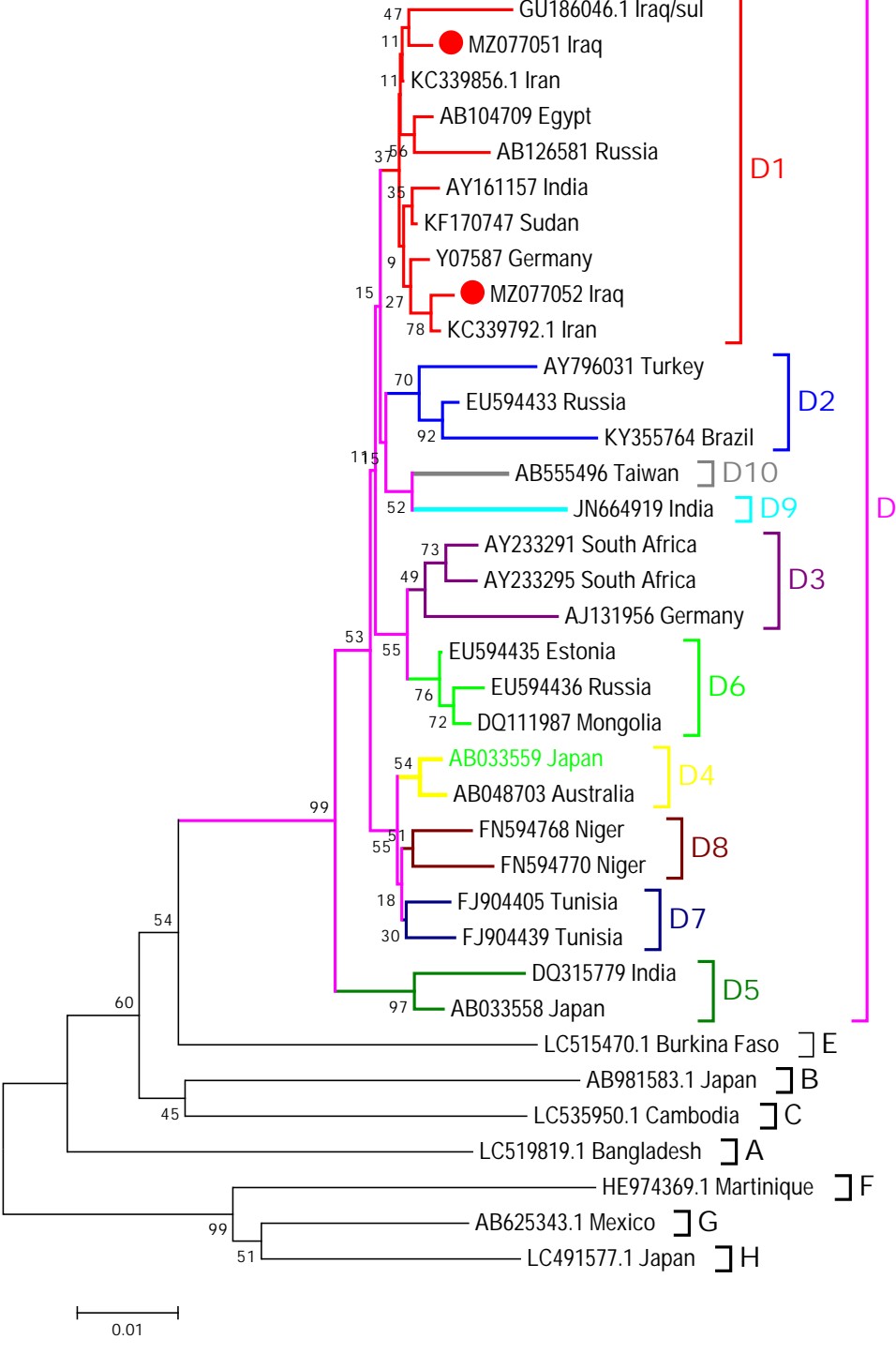

**Figure 3** **Phylogenetic tree of HBV nucleotide sequences of pres1/preS2 and S gene estimated with the neighbor-joining algorithm using MEGA version 10.** The topology was supported by bootstrap analysis with 1,000 replicates. Sulaimani HBV sequences reported in this study are represented as red circles. Sulaimani HBV was clustered in subgenotype D1.

codon at residue 69 of the S gene. In the S gene's amino acid residues 99 to 169, only one P113S mutation was detected in the middle hydrophilic region (MHR). To our knowledge, no substitution in the amino acid residue range from 124–147 is responsible for the determinant "a." (Fig. 2).

## DISCUSSION

The HBV infection is not uncommon in Iraq; the prevalence of this viral infection was estimated between 0.67–1.37%, according to some research in Iraq (*Hussein et al., 2021*; *Othman & Abbas, 2020*; *Babanejad et al., 2016*). However, there was an underestimation of the HBV prevalence in Iraq. Most studies detected only HBsAg as a marker of prevalence and did not measure the total anti-Hepatitis B core antibodies as an additional marker. The HBV infection can become chronic and seriously threaten physical, mental, and social health. As a result, the Iraqi Ministry of Health implemented a free HBV vaccination for all newborns in Iraq (*Hussein, 2018*). Unfortunately, little is known about the circulating genotypes of HBV in Iraqi populations, particularly in Iraq's Kurdistan region.

We analyzed the HBV genotype distribution among 33 chronically HBV-infected patients in the current study. The genetic analysis of HBV genotypes revealed the presence of only one genotype out of six genotypes investigated in the Sulaimaniyah governorate. All the patients were positive for only genotype D. Our results differ from those of previous Iraqi studies; in 2013, researchers genotyped four HBV-positive specimens in Sulaimaniyah city and found mixed genotypes (A+B+C+D) in all of the samples (*Rashid & Salih, 2015*). In addition, a further study discovered six distinct genotypes (A-F) with varying degrees of mixed infection prevalent among Iraqi HBV patients in the Baghdad Province of Iraq (*Mohsen, Al-azzawi & Ad'hiah, 2019*). Further analysis in Iraq's Samaara provinces identified a single infection with HBV genotypes A-F and a mixed infection with HBV genotypes C and F (*Abdulrazaq & AL-Azaawie, 2017*). On the other hand, our present analysis is consistent with previous research in the Duhok province Kurdistan region, which found that genotype D was the most common HBV genotype with no mixed infection (*Abdulla & Goreal, 2016*). Globally, genotype D was found in nearly 22.1% of the entirely HBV-infected people, of which 61.9% were found in Asia, with 22% in Africa and 13.5% in Europe (*Velkov et al., 2018*).

Phylogenetic analysis showed that the HBV sequence belongs to sub-genotype D 1/serotype aw, subtype ayw2. In North Iran, researchers studied 100 patients (*Moradi et al., 2012*) and 24 samples from Tehran, Iran (*Moradi et al., 2012*), and found comparable results. Therefore, these results may be due to the circulation of D1/ayw2 genotypes in the neighboring countries (North Iraq and North Iran).

One of the HBV samples examined in this investigation (HBV/Sul-2/2021) contained two deletion mutations in the Pre-S1-S region that were relatively long deletions. HBV/Sul-2/2021 accession number MZ077052 was closely identical to Iran HBV isolates KC339792 (*Karlsen et al., 2022*), raising the issue of whether this strain is created locally in Sulaimaniyah City owing to local mutation or it is brought from outside the city and even country. Another isolate, HBV/Sul-1/2021 accession number MZ077051 was found to be

closely related to the GU186046 isolate from 2013 (*Glebe et al., 2021*). This long deletion mutation may partly explain the prevalence of several HBV genotypes and subtypes; as a result, the virus is vulnerable to frequent mutations. In addition, deletion mutations may be immune escape mutations (*24, 2016*) and may be linked to decreased HBsAg expression (*Wang et al., 2018*). PreS deletion in carboxylic terminus mutants was shown to be associated with an increased risk of hepatocellular carcinoma development in prior research (*Chen et al., 2007*).

### Genetic analysis of preS1/preS2 region

The Sulaimaniyah HBV isolates in the current investigation shared a deletion of the first 11 amino acids of the Pre-S1 region, which is common in genotype D (*Hadad et al., 2018*). Point mutations at M88L and T90A, and D114N at the carboxylic end of PreS1 of HBV/Sul-1/2021 may alter viral pathogenicity and worsen disease development by increasing liver damage as well as HBV viral load. Several mutations (T26P, V55A, and V88L) in the Pre-S1 region of HBV/Sul-2/2021 need to be studied further to see how these mutations affect protein expression and the severity of the illness. G2765A substitution in the Pre-S1 of HBV genotype C was identified by Ogura et al., which resulted in lower L protein production and low viral load in CHB patients (*Ogura et al., 2019*). The limitations of this study were the small sample size and the lack of control for all HBV genotypes, but we attempted to address the latter limitation by sequencing two samples.

## CONCLUSIONS

According to this study, patients with chronic HBV infection seem more likely to have the D1 sub-genotype. The HBV isolates with the accession number MZ077052, named HBV/Sul-1/2021, were shown to have a novel HBV sequence with long deletion mutations. The second isolate, HBV/Sul-2/2021, has a few substitution mutations in the Pre-S1 region and S promoter. The ayw2 sub serotype of the HBV genotype was identified in this study. Extensive research is required in the future to fully understand the genotypes and mutations of HBV in chronically infected patients.

## ACKNOWLEDGEMENTS

The authors would like to thank the healthcare staff from the Shahid Hadi Consultation Clinic, Sulaimaniyah city, Iraq, for their kind help and support of this study.

### Funding

The authors received no funding for this work.

### Competing Interests

The authors declare there are no competing interests.

## Author Contributions

- Mardin Othman Abdulqadir conceived and designed the experiments, analyzed the data, prepared figures and/or tables, authored or reviewed drafts of the article, and approved the final draft.
- Peshnyar Muhammad Atta Rashid conceived and designed the experiments, performed the experiments, prepared figures and/or tables, and approved the final draft.
- Ali Hattem Hussain conceived and designed the experiments, performed the experiments, authored or reviewed drafts of the article, and approved the final draft.
- Heshu Sulaiman Rahman performed the experiments, analyzed the data, prepared figures and/or tables, authored or reviewed drafts of the article, and approved the final draft.
- Shahow Abdulrehman Ezzaddin analyzed the data, prepared figures and/or tables, and approved the final draft.

## Human Ethics

The following information was supplied relating to ethical approvals (i.e., approving body and any reference numbers):

The Kurdistan Institute for Strategic Study and Scientific Research approved the study (MLD38-2020).

## DNA Deposition

The following information was supplied regarding the deposition of DNA sequences:

The sequences are available at GenBank:
- HBV/Sul-1/2021: MZ077051;
- HBV/Sul-2/2021: MZ077052.

## Data Availability

The raw data is available in the Supplemental Files.

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
