# Peer review of "Genetic characterization of hepatitis B virus genotypes among patients with chronic infection in Sulaimaniyah city, Iraq"

_PeerJ, doi:10.7717/peerj.14454_

## Round 0.1 · original submission · Major Revisions

There are major issues in the manuscript, in particular, those pointed out by Reviewer 4. Please also make use of a professional English language copywriter since there are also many issues in that area.

·

Basic reporting

The article needs to review language
Most literature references are old. It needs to update the information.
Methodology and the structure are fair.
Table 2 is not a table. it's a figure taken from the alignment software. This table is not clear.

Experimental design

The scope of the paper is fair.
Technical methods is not new and there is no novel in the experimental work.

Validity of the findings

Detect the mutations is fine for this manuscript as new findings.
conclusion is fair

Additional comments

This manuscript needs more revised by author.
change table 2 because it is not clear.

Reviewer 2 ·

Basic reporting

The language in this manuscript can improve and some citations are missed through the work. Authors need to support some previous observations to give sense to this research. On the other hand, there are some cites that come from unsuitable references and authors must amend this issues. Moreover, I encourage to the authors to share more raw date to validate the findings here reported. Finally, I suggest a change in the title for this work "Genetic characterization of hepatitis B virus genotypes among patients with chronic infection in Sulaimaniyah, Iraq

Experimental design

I suggest authors to include more data on sequencing methods as it is the instrument used by this company, and if possible, some quality parameters for sequencing (e.g. depth coverage). On the other hand, a final comment on Nested PCR methods should be added: when authors declare that this multiplex nested PCR was carried out by adding “10 pmol of each primer”; I assume that in the second round, outer and inner primers has been used since only reverse or forward primers are reported in Table 1 for each genotype. This approach resembles a semi-nested asymmetrical PCR and if this is the case, it should be cleared in the manuscript. Also, it is necessary to declare in the manuscript what kind of positive control was used for these PCR reactions and how this control was characterized (if not a standard or reference material).

Validity of the findings

I think that authors should conclude on the scope for this research, taking in to account the limitations for the experimental design of this research. For example, when they declare “In Sulaimaniyah city, The HBV genotype belongs to ayw serotype/subtype ayw2”, this cannot be assured since two viral genotypes is not enough to generalize, but it is correct if they declare as “recorded” or “observed genotypes” in this work and for this region. Moreover, I encourage authors to share the band pattern for each of these 33 patients (as raw data), to undoubtedly validate the observation that only Genotype D has occasioned all the chronic HBV disease cases in Sulaimaniyah Governorate

Additional comments

No comment

Annotated reviews are not available for download in order to protect the identity of reviewers who chose to remain anonymous.

·

Basic reporting

According to my opinion, the article is good and suitable for publication.
and 'no comment'

Experimental design

The article is valid for publication,
and 'no comment'

Validity of the findings

Figures and tables are acceptable.
no comment.

Additional comments

The article is a good and documenting local hepatitis B virus strains is an important step.

Reviewer 4 ·

Basic reporting

The authors should strongly improve the English language to improve its understanding. For example, the lines 48 to 51 are scientifically just wrong (viral replication requires the RTase, evolution of the virus, therefore, cannot occur in “absence” of the enzyme), I would like to believe, because of difficulties in expressing the idea in a correct way. Similarly, lines 154, 159, and 160 are formulated for example in a rather unfamiliar way. It stays unclear what is meant for example on lines 221 and 222. To improve substantially the writing style, grammar, and readability, I suggest getting help from a colleague who is proficient in English and familiar with the topic of your article or contracting a professional service.
Improving the structure of the introduction would strengthen the goal of the study: to determine the prevalence of HBV genotypes in Sulaiminyah. It would be for example of interest to contrast already within the introduction the contradictory results of previous studies performed in Sulaimaniyah, Theran, Wasit, and northern Iran (refs. 11, 12, 14, 15). Presenting these results stresses the need for a broader, better HBV typing study in the given region to clarify these apparent discrepancies or differences.
The nomenclature used to describe the mutations is not standard. For mutations described at the nucleic acid level the nucleotide position is followed directly by the original base, a mayor sign ">" and the substituting base, example 2765G>A. In case of substitutions at the amino acid level the mutations are described by the amino acid position flanked by the original and the substituting amino acids. Furthermore, to avoid confusions, it is recommended to use the three-letter code, for example: Thr26Pro.

When referring to a mutation in a promoter region, it makes no sense to use the amino acid notation, even though in this case the promoter region lies within the coding region of an upstream gene.

It is unclear why on lines 249 and 250 “two” mutations are mentioned, but three are listed.

As it wasn’t possible to me to understand the claims in lines 48 to 51 the reference 4 seems to be wrongly cited as it discusses the role/mechanism of splicing rather than evolutionary mechanisms.

With respect to the figures there are four mayor and one minor issues (in this order):
1. The authors performed a phylogenetic analysis, it is, however, not shown. They should include the tree, I would suggest, together with the sequence alignment.
2. The quality of the sequence alignment labeled as table 2 is quite poor. There are more than 20 complete genomes available for HBV genotype D dated prior to the publication time of the HBV/Sul-1/2021 sequence. The authors should update the figure including all available HBV-D sequences and improve its quality using standard free programs like https://msa.biojs.net/, https://www.jalview.org/, https://www.ncbi.nlm.nih.gov/projects/msaviewer/, https://alignmentviewer.org/, https://www.ebi.ac.uk/Tools/msa/mview/, or others. It is rather a figure than a table.
3. Table 1 lists the general and type-specific primers for the nested multiplex approach, they contain, however, various errors in comparison to the original corrected publication. The table footer refers to bases M and Y, these degenerate bases don’t appear in the mentioned sequences.
4. Figure 1 should include positive controls for other genotypes, at least for those previously reported in the region and surroundings.
5. Figure 1 mentions the use of a size standard, there are, however, numerous different "100 bp ladders" available on the market. The authors should identify at least some of the fragment sizes next to the ladder.

Experimental design

The study was performed to detect the genotypes present in HBV infected patients of the Sulaimaniyah governorate. As previous studies were performed with the same primer sets, however, in a technically non-satisfactory manner (ref. 11) hampered with non-specific amplifications that weren’t addressed neither in that publication nor in the current manuscript it is quite compelling and meaningful to perform the current study. Technically it has been performed on a higher standard (no non-specific bands are visible), although no positive controls for genotypes different than HBV-D have been included or shown.
While the methods are described in good detail, the mayor reagents, i.e., the primers were not. As the primers are directly derived from another publication the corresponding table 1 may be omitted as it does not add to the replicability of the study.
On the other hand, the phylogenetic tree has not been included in the article, I may suppose it is the missing “figure 2” referenced on line 124 and again on line 167.

Validity of the findings

There is a benefit, at least locally, of the current study. Although it is rather small, based on the analysis of 33 HBV-positive patients, it corrects a previous study, documenting now the so far exclusive presence of genotype D in this region as it would be expected according to data available in the literature (for example: Velkov S, et al. Genes (Basel). 2018;9(10):495). It is important to contrast the current results and especially analyze critically the previous results available for the region of Sulaimaniyah (reference 11!).
It is, however, confusing to read in line 123 that 36 HBV sequences have been used for the construction of a phylogenetic tree, while only 33 patients were included in the study and only two of these strains were sequenced. Which are the other 34 sequences? Line 163 adds to the confusion as it states that 10 bp were used for the phylogenetic tree. It surely is a typo, but it constructs a picture of a manuscript that has been prepared quite sloppily.
Finally, the concluding remarks suffer as the rest of the manuscript from unclear phrasing. Pre-S1 as well as pre-S2 deletion mutants have been found in different genotypic backgrounds. The authors state that “genotype D is subject to deletion mutants” which creates the wrong impression that the detection of the deletions in genotype D is novelty.

---

## Round 0.2 · Major Revisions

Besides the extensive comments by the reviewers, please verify the sources of reference 16-20, since they are from predatory publishers and it is questionable by the reviewer.

Reviewer 2 ·

Basic reporting

Major changes have been done to the Introduction section, writing is much better, and objective of this research is implicit at the end of this section. Most of my previous notes have been attended (in this section) due to these changes. I think, it is not necessarily to give excessive details about HBV prevalence; worldwide and Iraq’s prevalence would be enough. Cite [10] is much better and belong to a suitable journal.

The reference previously cited in Line 58, was just removed and no other reference to support this statement was provided.

With respect to the references signaled previously in Discussion section, authors have updated reference [10], but reference [16] remains in the same status. Additionally, authors have cited references [17], [18], [19] and [20], which are articles from unsuitable publishers and journals (see more in my notes). I discourage citing these works, and authors must use more suitable references to discuss on this topic.

Molecular weight marks were added in Figure 1 for helping readers to better appreciate results. The note about positive control is suitable for this case. Table 2 has been also corrected and adapted for showing the most important findings. A Figure 3 has been added to this work which is very ad hoc for this work objectives. Nonetheless, I think that Figure 3 need some corrections for improving the quality of such image. For example, some numbers are superposed, and shadows can be noticed in square brackets and curly braces. Also, some country names need to be corrected (e.g. Bangladesh, México and South Africa). Moreover, it is desirable that Figure 3 had an indicator of the red dot meaning (which is: “reported in this study”) next to the accession numbers MZ077051 and MZ077052.

Besides, no cited document supporting such categorization for Genotype D by 11 amino acids deletion identification was given in line 156.

Experimental design

When describing “Subjects” section, mainly in studies including human being patients; it is desirable to make explicit in detail the inclusion and exclusion criteria. If these are the unique criteria, just add a sentence explaining which patients where not considered for this study. Please, consider giving at least the method and brand of HBsAg assay for this analyte detection in patients. Additionally, I insist that authors should include more details on blood sample collection and preprocessing of coagulant blood samples.

Regarding the nested PCR-method used in this research, I have been explained in detail by authors about the primers and type of analysis, and it is clearer than before. Further, the manuscript and Table 1 have been corrected and describe better for readers understanding.

Validity of the findings

The conclusion is better written, concrete and more focused on the scope of this work. Also, authors implicitly recognize limitations of this study, by leaving the door open for future research to corroborate these findings and allowing new hypotheses.

Additional comments

I have noticed that some of my previous suggestions were not attended. Please, I beg the authors to reconsider them (if applicable after changes in the manuscript), if they think it would improve the quality of this manuscript.

Annotated reviews are not available for download in order to protect the identity of reviewers who chose to remain anonymous.

Reviewer 4 ·

Basic reporting

English grammar and style has been improved, however, it isn't yet satisfactorily. I'll attach the PDF with lots of annotations as examples where the text must be improved in order to be clear and understandable. These annotations are, however, not comprehensive.
Figures must still be improved:
Figure 1 (or is it figure 3?, numbering is strange) is now truncated and the labeling of the sizes is sloppy.
Figure 2 contains a lot of typos and even conflicting data. Sequence AJ131956 Germany is listed twice, once as D3 and once as D5???
I insist in the correct naming of the mutations. If you refer to mutations at the DNA level it is not possible to refer to an amino acid sequence even though there are various overlapping genes in this region.
When mentioning the genotype prevalence in Iraq, there is at least one further recent article that should be cited: Dawod Salman A, Abbas Ali I, Haseeb Hwaid A. Prevalence of occult hepatitis B infection in Diyala province, Iraq. jidhealth [Internet]. 2022;5(2):685-92.
Previous data should be furthermore discussed critically (including quality of the data), inasmuch they are contradicting the findings in this manuscript. It is a pity in this context, that the authors do not have controls for other genotypes. This must be mentioned as a limitation of this study.

Experimental design

The lack of positive controls for the genotypes other than D is a very weak point. You should get into touch with other researchers in order to obtain these controls.
Ali H. Ad'hiah of the Tropical-Biological Research Unit, College of Science, University of Baghdad, published recently an article claiming of having detected genotypes A, B, C, D, E, and F.
(dr.ahadhiah@sc.uobaghdad.edu.iq)
Of course, this article adds to the information about recently found HBV genotypes in Iraq:

Mohsen RT, Al-azzawi RH, Ad'hiah AH. Hepatitis B virus genotypes among chronic hepatitis B patients from Baghdad, Iraq and their impact on liver function. Gene Reports, 2019. 17:100548. DOI:10.1016/j.genrep.2019.100548.

Validity of the findings

The lack of positive controls for detecting genotypes other than D must be mentioned. They limit the observation that no other genotype has been detected. Any conclusion about the exclusive detection of HBV genotype D must be removed or must be put into perspective.

Additional comments

The article still does not meet a minimum standard in order to be publishable.
The lack of controls should be addressed.
The efforts to improve the quality of the manuscript should be increased.

Annotated reviews are not available for download in order to protect the identity of reviewers who chose to remain anonymous.

---

## Round 0.3 · Minor Revisions

Please take into considerations the detailed comments by the reviewer. Best regards.

Reviewer 2 ·

Basic reporting

The use of English language is much better since the first draft, citation and use of literature references have improved in quality and figures have also good quality. Reference 19 needs style correction as it is required for the journal. Most of my previous observation have been attended.

Experimental design

Authors have already included inclusion criteria, and details about analysis and blood sample procedures, which I consider a very positive complement for this research. I have no additional comments about this section.

Validity of the findings

I see that authors added a final clause of limitations of this study, which is something very positive for any type of academic paper. Conclusion is very concrete and ad hoc for the findings here obtained. I have no additional comments for this section

Additional comments

No comments

Reviewer 4 ·

Basic reporting

For the last revision, I wrote:
When mentioning the genotype prevalence in Iraq, there is at least one further recent article that should be cited: Dawod Salman A, Abbas Ali I, Haseeb Hwaid A. Prevalence of occult hepatitis B infection in Diyala province, Iraq. jidhealth [Internet]. 2022;5(2):685-92.
You answered:
Dear reviewer, this article not mention any genotyping, we claimed using sequencing and phylogenic analysis but there is no any result about these data because they got negative result by PCR. Based on your advice, we used latest reference about HBV prevalence in Iraq (Reference 4, lines 38-39, grey color). Thanks

You are absolutely right. I apologize.

However, there are still a lot of changes that are required to improve the readability, English grammar, style and orthography as well as to correct errors in presented data and scientific expression:

Line 27
A promoter region is by definition a DNA sequence (https://www.genome.gov/genetics-glossary/Promoter)
Therefore, it is NOT possible to delete "amino acids" from a promoter region. A promoter is a DNA element not a peptide, mutations referring to it must be named accordingly.
Although, as you state in your letter, this region is comprised in three different genes read in different frames you still refer here to a promoter and thus to a DNA sequence.
Furthermore, there are recommendations about the naming of mutations: den Dunnen JT, Dalgleish R, Maglott DR, Hart RK, Greenblatt MS, McGowan-Jordan J, Roux AF, Smith T, Antonarakis SE, Taschner PE. HGVS Recommendations for the Description of Sequence Variants: 2016 Update. Hum Mutat. 2016 Jun;37(6):564-9. doi: 10.1002/humu.22981.
While they were originally developed for naming mutations in human sequences, the recommendations apply to all species.

The following article uses for example the three-letter code nomenclature for naming HBV mutants:
El Hadad S, Alakilli S, Rabah S, Sabir J. Sequence analysis of sub-genotype D hepatitis B surface antigens isolated from Jeddah, Saudi Arabia. Saudi J Biol Sci. 2018 May;25(4):838-847.

You may further read about the rules of the above mentioned nomenclature at:
http://varnomen.hgvs.org/

Line 54
It is not the antiviral drug that REACTS differently TO various genotypes but
either
The antiviral drug that ACTS differently ON various genotypes
or
Various genotypes RESPOND differently TO antiviral drugs

Take for example the title of the cited article 9:
"The response of hepatitis B virus genotype to interferon is associated with a mutation in the interferon-stimulated response element."
As stated above: the genotype responds to the treatment. It is not the other way around.
Correct please.

Line 113
CLOSED is the opposite of OPEN. (https://www.oxfordlearnersdictionaries.com/us/definition/english/closed)
What is a CLOSED PERCENTAGE? Please correct this expression.

Line 129
PCR uses at least two primerS or a primer PAIR. Please correct.

Line 164
There exist nonsense, missense, and silent or synonymous mutations.
(https://www.genome.gov/genetics-glossary/Point-Mutation)
(https://bmccancer.biomedcentral.com/articles/10.1186/s12885-019-5572-x)
What are "sense point mutation divergences"?
Correct please!

Line 174
Correct the typo: substation -> substitution

Lines 207 to 217 (block A) and lines 219 to 230 (block B)
Both blocks of text are sentence by sentence just a different wording of the same discussion. Two examples (first and last lines)
A:
One of the HBV samples examined in this investigation (HBV/Sul-2/2021) contained two deletion mutations in the pre-S1-S region that were relatively long deletions.
B:
In the current study, one HBV out of the 33 specimens (HBV/Sul-2/2021) had relatively long two deletion mutations in the pre-S1-S locus.

A:
PreS deletion in carboxylic terminus mutants was shown to be associated with an increased risk of hepatocellular carcinoma development in prior research [19].
B:
Furthermore, data from a previous study demonstrated the prognostic significance of preS deletion in carboxylic terminus mutants in developing hepatocellular carcinoma and liver cirrhosis [26].

Remove one of these blocks!

Lines 207 to 217 (block A) and lines 219 to 230 (block B)
In addition, the citations therein point to different references:
19. Abdulrazaq, G. and A.F. AL-Azaawie, Molecular and Immunological Study of Hepatitis B virus infection in Samara City, Iraq. 2017.
26. Chen, C.H., et al., Pre-S deletion and complex mutations of hepatitis B virus related to advanced liver disease in HBeAg-negative patients. Gastroenterology, 2007. 133(5): p.1466-1474.

Do you mean reference 19 or 26 in this context?
Correct the citations in the text block A o B that you are going to keep.


Lines 282, 284, 299, 307
References are incomplete.
Correct please.

Line 343
Change indicted to indicated

Figure 3
The following sequences do not point to the correct country:
KF170747, EU594436, AB033559, FN594768, FN594770, LC515470, LC535950
Correct the figure.

Experimental design

no comment

Validity of the findings

no comment

Additional comments

no comment

---

## Round 0.4 · accepted · Accept

Thanks for the careful revision for this final version.